# Psychological distress, resettlement stress, and lower school engagement among Arabic-speaking refugee parents in Sydney, Australia: A cross-sectional cohort study

**Jess R. Baker**\*, **Derrick Silove, Deserae Horswood**, **Afaf Al-Shammari**, **Mohammed Mohsin**, **Susan Rees**, **Valsamma Eapen**

School of Psychiatry, University of New South Wales, Sydney, Australia

\* Jessica.baker@unsw.edu.au

## Abstract

### Background

Schools play a key role in supporting the well-being and resettlement of refugee children, and parental engagement with the school may be a critical factor in the process. Many resettlement countries have policies in place to support refugee parents' engagement with their children's school. However, the impact of these programs lacks systematic evaluation. This study first aimed to validate self-report measures of parental school engagement developed specifically for the refugee context, and second, to identify parent characteristics associated with school engagement, so as to help tailor support to families most in need.

### Methods and findings

The report utilises 2016 baseline data of a cohort study of 233 Arabic-speaking parents (77% response rate) of 10- to 12-year-old schoolchildren from refugee backgrounds across 5 schools in Sydney, Australia. Most participants were born in Iraq (81%) or Syria (11%), and only 25% spoke English well to very well. Participants' mean age was 40 years old, and 83% were female. Confirmatory factor analyses were run on provisional item sets identified from a literature review and separate qualitative study. The findings informed the development of 4 self-report tools assessing parent engagement with the school and school community, school belonging, and quality of the relationship with the schools' bilingual cultural broker. Cronbach alpha and Pearson correlations with an established Teacher–Home Communication subscale demonstrated adequate reliability (α = 0.67 to 0.80) and construct and convergent validity of the measures (*p* < 0.01), respectively.

Parent characteristics were entered into respective least absolute shrinkage and selection operator (LASSO) regression analyses. The degree of parents' psychological distress (as measured by the Kessler10 self-report instrument) and postmigration living difficulties (PLMDs) were each associated with lower school engagement and belonging, whereas less time lived in Australia, lower education levels, and an unemployed status were associated with higher ratings in relationship quality with the schools' cultural broker. Study limitations

**Data Availability Statement:** The data that support the findings of this study will be made available to researchers who meet the criteria for access to study data, after permission from the relevant ethics body below has been sought. The Human

Ethics Coordinator The University of New South Wales Email: humanethics@unsw.edu.au.

**Funding:** This research was part funded by the National Health and Medical Research Council of Australia (NHMRC) program grant (APP1073041) awarded to DS (CI) and SR (AI). The NHMRC had no role in study design, data collection and analysis, decision to publish, or preparation of the manuscript.

**Competing interests:** I have read the journal's policy and the authors of this manuscript declare that no competing interests exists.

**Abbreviations:** CALD, culturally and linguistically diverse; CBRS, Cultural Broker Relationship Scale; CFA, confirmatory factor analysis; CFI, comparative fit index; EAD/L, English as an Additional Language or Dialect; HTQ, Harvard Trauma Questionnaire; K10, Kessler Psychological Distress Scale; LASSO, least absolute shrinkage and selection operator; PMLD, postmigration living difficulty; PRESS, predicted residual sum of squares; PSSM, Psychological Sense of School Membership; PTE, potentially traumatic event; SBS-RP, School Belonging Scale-Refugee Parent; SCES-RP, School Community Engagement Scale-Refugee Parent; SIES-RP, School Internal Engagement Scale-Refugee Parent; SRMR, standardised root mean square residual; TE, traumatic event; THC, Teacher–Home Communication; TLI, Tucker–Lewis index; VIF, variance inflation factor.

include the cross-sectional design and the modest amount of variance (8% to 22%) accounted for by the regression models.

## Conclusions

The study offers preliminary refugee-specific measures of parental school engagement. It is expected they will provide a resource for evaluating efforts to support the integration of refugee families into schools. The findings support the need for initiatives that identify and support parents with school-attending children from refugee backgrounds who are experiencing psychological distress or resettlement stressors. At the school level, the findings suggest that cultural brokers may be effective in targeting newly arrived families.

## Author summary

### Why was this study done?

- Current measures of parent school engagement do not consider the unique experiences of families from refugee backgrounds.

- This study was undertaken to develop appropriate measures of refugee parents' engagement with school.

- It is expected that specific measures to ascertain refugee parents' engagement will help support the development and evaluation of school-based refugee programs.

### What did the researchers do and find?

- Using data from a cohort study of 233 Arabic-speaking refugee families, the study developed and validated 4 self-reports tools to assess parent school engagement and their sense of belonging.

- The study also explored the quality of the relationship between parents and the bilingual officer employed by schools to act as a cultural broker.

- Our findings suggest that parents who were experiencing psychological distress or post-migration living difficulties (PLMDs) were less engaged with their child's school.

### What do these findings mean?

- These new refugee-specific measures of parent engagement are expected to assist in evaluating programs aimed at supporting the integration of refugee parents and their children into schools.

- The cross-sectional design precludes causal interpretations.

- Nonetheless, the findings highlight a need to consider the identification and capacity building of parents who are experiencing psychological distress or resettlement stressors.

## Introduction

Global refugee numbers are at a record high, of which school aged children make up over 52% [1]. In high-income countries of resettlement, school attendance is mandatory. This provides an opportunity to engage with parents from refugee backgrounds, in order to promote the well-being and community integration of refugee families as a whole [2–4]. In that regard, refugee parents' relationship with the school may be a critical factor in the successful adaptation of refugee students [5,6]. Robust evidence suggests that across ethnic groups and sociodemographic backgrounds, parent engagement with schools is positively correlated with children's well-being. More specifically, children whose parents participate in school activities are more engaged in their learning and more successful in terms of achievement, attendance, motivation, school completion, emotional adjustment, prosocial behaviour, and peer interactions [7–14].

Several countries such as the United Kingdom, United States of America, Canada, Australia, Sweden, and Germany have government policies and programs in place to support the integration of refugee families into schools. Targeted measures that capture the distinct ways in which refugee parents might engage with schools are needed to monitor the success of these initiatives. Specialised measures may also generate data to better advance efforts to improve refugee parents' engagement in their children's schooling in a tailored way that is most valuable to the family. Teasing out the nuances of refugee parent school engagement relative to non-refugee families has the additional benefit of avoiding any cultural biases, including misguided findings that refugee families might be less invested in their children's education. Several theories of parent school engagement have been proposed [15–17]. The most common and comprehensive framework proposed is Epstein's 6 different types of parent engagement [16]. The first type of engagement is parenting and discusses the school's role in educating families on how to create a positive home environment. The second type of engagement is centered on effective communication between home and school. The other types include recruiting parents to volunteer at the school, involving parents in school decision-making, home learning, and community collaborations.

These traditional definitions of parental school involvement may not adequately capture engagement as it applies to refugee families [18]. Refugee parents' limited English proficiency [19–23] may interfere with volunteering at the school and supporting their child's learning at home. This issue may be exacerbated by time constraints related to work, family, and economic survival [19,20,23,24]. Disruptions in schooling during displacement and limited family experience with school systems in countries of resettlement may also challenge typical parent engagement [24,25]. Qualitative research also suggests cultural dissonance between expectations of parents and the school. For instance, culture of origin may influence refugee parents' belief that it is inappropriate to "interfere" with the work of teachers and that keeping one's distance may be deemed to show respect for the school and its staff [26].

Indeed, the high value refugee parents place on their child's education may be expressed in different ways [23]. While not focused on refugee children, a large national study of students in the US found that academic achievement among white American children was positively associated with parent engagement in school-based activities; however, for Asian American families, the relationship was inverse [27]. In another study, ethnicity was found to influence the relationship between different types of parent engagement categorised within Epstein's typology and student achievement [28]. For instance, volunteering was a better predictor of achievement for white students than for Asian, Hispanic, and black students, while contact with the school was more important for explaining achievement effects for Hispanic and white students than for Asian and black students.

Moreover, an emphasis on behavioural indicators of parent engagement [15–17] may undervalue the relational or more affective components of parental engagement, such as how close or connected a parent feels to the school or school community. The well-being of refugee parents is affected by historical factors such as past trauma and displacement [29] and learned distrust of authorities based on exposure to persecution and discrimination [30]. Establishing parental trust and safety may therefore be foundational to effective engagement with schools. For example, in a study of Hmong refugees in the US, a school ethos of inclusion was found to assist parents to overcome expectations of deference towards "authority" [23].

Numerous case studies have showcased how adept certain schools and their staff have become at engaging with refugee families [31,32]. Current measures of parent school engagement may not capture these specific activities attuned to refugee families [33]. For instance, schools have been found to play an instrumental role in guiding parents to local support agencies and helping them navigate unfamiliar and often overwhelming government systems [31,34]. Similarly, reports of social activities at the school, such as breakfast clubs and diversity days, have been posited to connect parents with each other, while school-initiated parent classes to support acculturation in the region's primary language have been documented to improve cultural competency [35]. A key piece in the literature is the consistent role that bilingual school liaison officers play in facilitating these processes [6,23,26,36]. These persons are employed by the school, specifically tasked with brokering the relationship between refugee families and the school, as well as the wider community. To date, no measure exists to capture the quality of the relationship between the cultural broker and parents.

The primary aim of the present study was to develop and validate self-report tools to measure refugee parents' engagement with school. We draw on Epstein's framework and extend it to include specific elements of engagement that foster families' social and cultural capital [31,32]. We also aimed to develop and validate a novel self-report tool that includes an affective index of engagement—namely parents' sense of belonging or feelings of connectedness to the school. Further, we aimed to include an assessment of the relationship between refugee parents and the schools' cultural broker. An established measure of teacher–home communication, a robust facet of parent engagement, was selected as a distal measure of convergent validity [37–39].

The secondary aim of the present study was to identify refugee parent characteristics associated with these new measures of parent engagement. While some refugee families may engage readily and easily with school and not require special assistance, other families may have the opposite tendency and require more active assistance. Identifying refugee parents at "risk" of low levels of engagement can help schools to individualise or tailor their support to families most in need. It was reasoned that these factors would likely mirror some of the barriers to traditional engagement described earlier [19–24]. For instance, in the migrant literature, low socioeconomic status, less education, and time preoccupation with daily environmental stressors have been associated with lower parent engagement [40–46].

Specifically, it was hypothesised that countervailing parental factors such as length of time lived in the host country, employment and visa status, psychological health, English language proficiency, trauma experience, and postmigration stressors would be associated with school engagement and belonging.

## Method

### Design

The present report is based on the baseline cross-sectional data of a longitudinal cohort study exploring school climate and refugee student well-being.

## Ethics statement

The study was approved by the University's Human Research Ethics Committee (HC15833) and the equivalent body of the State Department of Education (SERAP 2016056) (see S1 Prospective Protocol). The bilingual research team was trained in working with refugee families using sensitive interviewing techniques and received ongoing supervision by author JB throughout the study. The principles governing recruitment were implemented according to the Strengthening the Reporting of Observational Studies in Epidemiology (STROBE) guidelines (see S1 STROBE Checklist).

## Setting

The study was conducted at 5 primary schools in a low socioeconomic multicultural geographical area of Sydney, Australia in 2016. The urban region receives a substantial portion of Australia's refugee intake each year. For example, over 12% of Australia's 73,833 refugee intake was settled in the catchment area between 2010 and 2015. An additional 12,000 Syrians were also resettled as a special intake in the area between 2016 and 2017 [47]. Given several years of experience working with refugee families, the schools had many well-established support initiatives, such as new family orientations, small support classes for refugee students, events to celebrate cultural diversity, homework clubs, English language and computer literacy classes for parents, and parent cafes and breakfast clubs.

## Participants

Refugees speaking Arabic were the largest group entering the country at the time. Confining the study to one participant language limited the risk of transcultural measurement error and small cell sizes that would eventuate if multiple ethnic groups were included. Participants were Arabic-speaking parents of Year 5 and 6 students comprising 10- to 12-year-old children, who entered Australia on a humanitarian (refugee) visa, and had attended the school for at least 3 months. Selection criteria permitted either mother or father to participate at the family's choosing. Assuming that the maximum standard deviation would not exceed 15 for the largest subscale total score for the project, and applying a 95% confidence interval along with a 3% margin of error, a sample size of 96 was deemed sufficient to reject the null hypothesis at the 0.05 probability level.

A nominated bilingual cultural broker at the school invited eligible families to participate, passing on the contact details of assenting families to the bilingual field team. At the point of further contact, the study process and objectives were explained fully in Arabic and, if the parent provided written informed consent, the family was included in the study (see flow diagram in Fig 1). Families received a $30 gift voucher for their participation. The design involved a rolling recruitment strategy of all eligible students attending or joining the school over a 2-year time frame, with the largest intake occurring at the start of each school year.

## Measures

Participants completed an extensive demographic survey followed by a provisional set of items for the school engagement measures (see a description hereafter), and then lastly, Arabic versions of the established measures below. All translations were checked for local cultural appropriateness and accuracy. To circumvent any literacy or comprehension issues, the measures were completed in an interview style, with the questions read aloud to participants in Arabic.

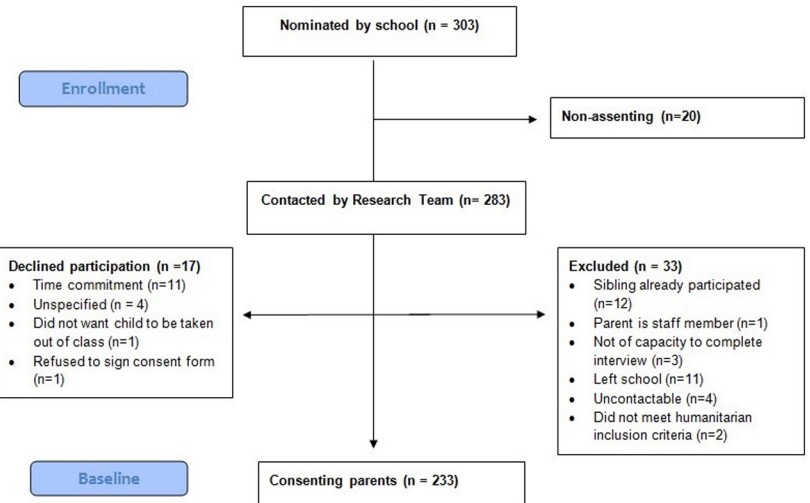

**Fig 1. Flow diagram of parent participant recruitment.**

## Item development and pilot for new measures

Items were developed from 2 main sources: (i) a desk review of the refugee school literature and existing school engagement scales [48,49]; and (ii) themes emerging from a preliminary qualitative study exploring the school experiences of refugee families (to be published elsewhere). The informing qualitative study included two 10 to 12 year olds (and their parents) from refugee backgrounds from each of the 5 schools of the present study, plus one other school that opted out before the quantitative stage of the current study ($n$ = 12 dyads). In addition, focus groups were held with staff at each school. These typically comprised a Year 6 Teacher, School Counsellor, an English as an Additional Language or Dialect (EAD/L) Teacher, and one bilingual cultural broker. In total, 26 staff members participated.

Based on the desk review and qualitative study, authors JB and DH generated a pool of potential items written in English aimed at measuring school engagement. These were reviewed and edited for content and cultural appropriateness by a local primary school teacher and author AA (an active member of the local Arabic-speaking community with excellent English language skills). Author AA translated the items into Arabic, which were assessed for comprehension and cultural congruence by Arabic members of the research team. The items were piloted with sequential parents enrolled in the study. Items were deleted if they proved difficult to understand, required excessive explanation, or were interpreted differently by parents. This iterative process, involving feedback and research group discussion, led to a final consensus regarding a provisional set of school engagement items, a provisional set of items pertaining to the cultural broker–parent relationship, and a provisional set of items pertaining to the parent sense of school belonging, as detailed below. The initial set of items can be found in supporting file Table A in S1 Data Tables.

## Refugee parent school engagement items

The 23-item measure followed 5 themes identified in the preliminary qualitative study (to be published elsewhere). The themes were (i) ease of communication between the school and parent (5 items, e.g., "It would be difficult for me to find a way to discuss an issue with school staff"); (ii) practical assistance provided by the school such as accessing services or provision of food or uniforms (5 items, e.g., "School staff direct me to services I need or that might be

helpful to access"); (iii) degree of parent participation in school activities (6 items, e.g., "I regularly attend school events"); and the ways in which the school facilitates (iv) social capital (4 items, e.g., "Through the school, I have met and made friends"); or (v) acculturation (3 items, e.g., "This school helps me feel like I belong in Australia"). To maintain consistency with the Teacher–Home Communication (THC) scale, items were rated on a 4-point Likert scale from 1 (disagree a lot) to 4 (agree a lot). Negatively worded items were reverse scored. Higher scores reflected greater school engagement.

## Cultural broker–parent relationship items

Six items pertained to the relationship between the parent and an identified bilingual cultural broker at the school employed to support newly arrived culturally and linguistically diverse (CALD) families in the transition to school and foster engagement between the school and broader CALD community [6,23,26]. Example items include: "XX has been important in making me feel secure in the school because xx understands my family's religious and cultural backgrounds" and "XX helps me with my family's wellbeing." Items were rated on a 4-item Likert scale. For parents unable to identify a cultural broker, items were coded as "Not Applicable" and excluded from any analysis. Higher scores reflected a higher-quality relationship with the cultural broker.

## Refugee parents' sense of school belonging items

Distinct to the measures above, parent sense of school belongingness items were based on an adaptation of the well-established student-rated Psychological Sense of School Membership (PSSM) scale [50]. Items measured feelings of belonging and engagement with the school and were rephrased to ensure relevance to parents (e.g., "I am treated with as much respect as other parents in the school"). Two irrelevant items concerning school performance were excluded. The resulting 16-item scale was rated and scored in accordance with the PSSM scoring procedure on a 5-point Likert scale (1 "Not at all true" to 5 "Completely true"). Negatively worded items were reverse scored. Higher scores reflected a greater sense of belonging.

## Established measures

**Teacher–Home Communication subscale of the Delaware School Climate Survey.** The Delaware School Climate Survey measures parent perceptions of the "quality and character of school life" across several domains utilising a 4-point Likert scale (1 "Disagree a lot" to 4 "Agree a lot"). For the study purpose, the 4-item THC subscale was summed, with higher scores reflecting more positive communication. The parent survey has shown excellent internal consistency across ethnicity, sex, and grade-level groups, and correlations with academic achievement, bullying victimisation, and school suspensions demonstrate concurrent validity [52]. The current sample demonstrated adequate internal consistency for the THC (Cronbach $\alpha = 0.69$).

**Kessler Psychological Distress Scale (K10).** The K10 is a 10-item self-report measure of nonspecific psychological distress for adults (e.g., "How often did you feel nervous?"). Participants rated their experience of distress in the preceding 30 days on a 5-point Likert scale (1 "None of the time" to 5 "All of the time"). Items were added to produce a total score; the higher the score, the greater the level of distress. Arabic NAATI-accredited translations were obtained from the state government health body. Normative data have been established for Australia [53], and high internal consistency demonstrated with refugee groups ($\alpha = 0.86$) [54]. The total score obtained from the present sample demonstrated excellent internal consistency (Cronbach $\alpha = 0.89$).

**Postmigration Living Difficulties Scale (PMLD).** The PMLD scale consisted of 28 items reflecting levels of distress caused by daily living difficulties related to postmigration adaptation over the previous 12 months (e.g., "separation from family" or "delays processing your visa application"). Items are rated on a 5-point scale (0: "not a problem" to 4: "a very serious problem"). The scale has consistently been identified as a predictor of psychopathology among displaced populations [55]. The Arabic translation was obtained with permission from the lead author of the scale [56]. The total PMLD counts were summed to create a continuous total score. Internal reliability for PMLD items for the study sample was 0.79.

**The Harvard Trauma Questionnaire (HTQ)—Iraqi version.** The HTQ is the most widely used measure of trauma exposure and posttraumatic stress disorder in the refugee field [57,58]. The initial section includes a checklist of potentially traumatic events (PTEs), including exposure to combat, forms of warfare, and separations and losses. The list was modified to reflect the experiences of the refugee populations being studied. Participants reported on their lifetime exposure to 23 events (Yes/No). Items were coded "1" for yes, and "0" for no for lifetime exposure. The items were summed to generate a total trauma count ranging from 0 to 23.

## Data analysis

**Refinement and psychometric testing of the new measures.** Confirmatory factor analysis (CFA) using MPLUS 7.1 was conducted on the 3 provisional measures above, with the aim to identify component scales and relevant item loadings. The threshold was set at $p < 0.05$. Items with lowest loadings were deleted serially to test for improved model fit based on a composite comparative fit index (CFI), Tucker–Lewis index (TLI), and standardised root mean square residual (SRMR) [51]. The final sets of items were aggregated, and transformations used to generate mean scores.

Reliability and validity tests were conducted using IBM SPSS Statistics Version 25 [59] and MPLUS 7.1 [51]. For reliability estimates, Cronbach alpha (α) was calculated for each scale. Positive correlations between items within each scale and with the single total score as indicated in a CFA were used to indicate construct validity. As an indicator of convergent validity, the data were examined for positive correlations among each derived scale, as well as the THC subscale.

**Predictive analyses with developed measures.** Firstly, descriptive statistics were presented for all parental level characteristics. Next, variance inflation factor (VIF) and tolerance statistics examined the multicollinearity among predictors accounting for each outcome factor. The original intention was to conduct linear or stepwise regression analyses. However, this was changed at the peer review stage, to LASSO (forward) regression analyses. Accordingly, parental level characteristics were entered into separate LASSO (forward) regression analyses with each newly developed scale as the outcome variable. The LASSO regressions were performed with SAS 9.4 software, PROC GLMSELECT with selection = forward (select = SL stop = predicted residual sum of squares (PRESS). To adjust for school effects, school was entered as a predictor variable. Based on the 5 sampled schools A, B, C, D, and E (designated letters for confidentiality), 4 indicators were created considering the fifth school (school E) as the reference group. Differences in measurement units of predictor variables meant that standardised LASSO regression coefficients were presented.

## Results

### Sample characteristics

The anticipated resettlement of 12,000 refugee families into the local area was delayed, thus the sample size was lower than prospective estimates. Of the 303 families identified by the schools

to take part, 283 families assented to be contacted by the research team, and a final 233 parents (77% of those identified) consented to participate—see flow diagram of parent participant recruitment in Fig 1. Parent characteristics are provided in Table 1. Most participants were married women, born in Iraq (consistent with Australia's refugee intake at the time), and spoke little or no English. Around a third were educated beyond high school, and few participants were currently employed. Around half met criteria for a mental health disorder based on the K10, and almost all endorsed experiencing at least one traumatic event (TE).

## Confirmatory factor analysis

Please see Fig 2 showing the process of measure development.

## Refugee parents school engagement

The CFA examined the correlations between the 5 predefined themes of communication between home and school, practical assistance provided by the school, parent participation in school activities, school as facilitating social capital, and school as enabling acculturation. The dimensions of communication, assistance, parent participation, and social capital correlated significantly with each other (r = 0.25 to 0.96, $p$-values ranged from 0.037 to 0.001). Acculturation did not show a correlation with the communication or assistance dimensions ($p > 0.0.05$) but significantly correlated with the social capital dimension (r = 0.68, $p < 0.001$) (see Table B in S1 Data Tables). Based on these findings, the items were aggregated into 2 notional and statistically distinct scales: (i) refugee parent engagement in relation to communication, assistance, and participation within the internal school system—named the School Internal Engagement Scale-Refugee Parent (SIES-RP) (16 items); and (ii) broader refugee parent engagement within the school community in which the school facilitates social capital and acculturation—named the School Community Engagement Scale-Refugee Parent (SCES-RP) (7 items). A CFA was performed on each separate scale. After item reduction as described above, the final CFA model fitness indices are presented in Table 2. Eleven of the original 16 items were retained for the SIES-RP, while all 7 items of the SCES-RP were retained (see Tables C and D in S1 Data Tables, respectively).

## Cultural broker relationship

Final CFA model fitness indices are presented in Table 2. Five of the original 6 items were retained. This was named the Cultural Broker Relationship Scale (CBRS) (see Table E in S1 Data Tables).

## Refugee parents' sense of school belonging

Final CFA model fitness indices are presented in Table 2. Fifteen of the original 16 items were retained. This was named the School Belonging Scale-Refugee Parent (SBS-RP) (see Table F in S1 Data Tables).

## Final measures

In summary, the refinement procedures produced 4 scales: the SIES-RP, the SCES-RP, the CBRS, and the SBS-RP. Cronbach α and other summary statistics are shown in Table 2.

## Reliability and validity testing of developed measures

Tables detailing the construct validity of the final developed measures are presented in Tables G-J in S1 Data Tables. Descriptive characteristics and Cronbach α (i.e., reliability) of

**Table 1. Parent characteristics.**

| Parents' characteristics | Number | % of total |
|---|---:|---|
| *All*[#] | *233* | *100.0* |
| **Sex** | | |
| Male | 39 | 16.7 |
| Female | 194 | 83.3 |
| **Parents' age** | | |
| <35 years | 62 | 27.0 |
| 35–44 | 108 | 47.0 |
| 45 and older | 60 | 26.0 |
| *Mean (SD)* | *39.7(6.8)* | |
| **Marital status** | | |
| Married | 201 | 86.3 |
| Divorced/Separated/Widowed | 32 | 13.7 |
| **Current parent visa category in Australia** | | |
| Humanitarian refugee/Family sponsored | 178 | 76.4 |
| Permanent resident or Citizen | 55 | 23.6 |
| **Duration of living in Australia** | | |
| Under 2 years | 66 | 30.1 |
| 2 to 5 years | 90 | 41.1 |
| More than 5 years | 63 | 28.8 |
| *Mean (SD)* | *3.8(3.0)* | |
| **Country of birth** | | |
| Iraq | 189 | 81.1 |
| Syria | 26 | 11.2 |
| Other | 18 | 7.7 |
| **Spoken language at home** | | |
| Arabic | 161 | 69.1 |
| Others | 72 | 30.9 |
| **English proficiency** | | |
| A little/Not at all | 174 | 74.7 |
| Well/Very Well | 59 | 25.3 |
| **Highest level of education attained** | | |
| Up to high school | 152 | 65.2 |
| Diploma and University degree | 81 | 34.8 |
| **Employment status** | | |
| Unemployed and others | 216 | 92.7 |
| Employed | 17 | 7.3 |
| **Family income (annual)** | | |
| up to $37,000 | 79 | 33.9 |
| $37,001 and above | 154 | 66.1 |
| **Parent received any counselling or psychological therapies in Australia** | | |
| No counselling or therapy | 129 | 58.1 |
| NGO service | 55 | 24.8 |
| GP | 10 | 4.5 |
| Private practice psychologist/counsellor | 25 | 11.3 |
| Others | 3 | 1.4 |
| ***Number TEs experienced:*** *Mean (SD)* | *9.9(4.0)* | |
| ***Number of PMLDs:*** *Mean (SD)* | *5.4(0.3)* | |

(*Continued*)

**Table 1.** (Continued)

| Parents' characteristics | Number | % of total |
|---|---:|---|
| *Psychological distress categories (measured as K10)*: Mean (SD) | *21.3(9.9)* | |
| **Child's enrolled school** | | |
| School A | 46 | 19.7 |
| School B | 43 | 18.5 |
| School C | 60 | 25.7 |
| School D | 33 | 14.2 |
| School E | 51 | 21.9 |

[#]The numbers do not always add up to 233 due to missing data. The number of missing observations for "Parents duration of living in Australia" was 14 (6%) and for "Parent received any counselling or psychological therapies in Australia" was 11 (4.7%). Else, the number of missing cases ranges from 0 to 3 (0 to 1.2).

GP, general practitioner; K10, Kessler Psychological Distress Scale; NGO, nongovernment organisation; PMLD, postmigration living difficulty; TE, traumatic event.

the new measures, as well as their correlations with THC, are provided in Tables 3 and 4, respectively. All measures were significantly correlated with THC indicating convergent validity. The SBS-RP alpha score also indicated very good reliability. The SCES-RP and SBS-RP reported alpha scores of a minimally acceptable standard. The Cronbach α of the SIES-RP and CBRS did not meet the typically acceptable standard of 0.70, reported as 0.67 and 0.63, respectively.

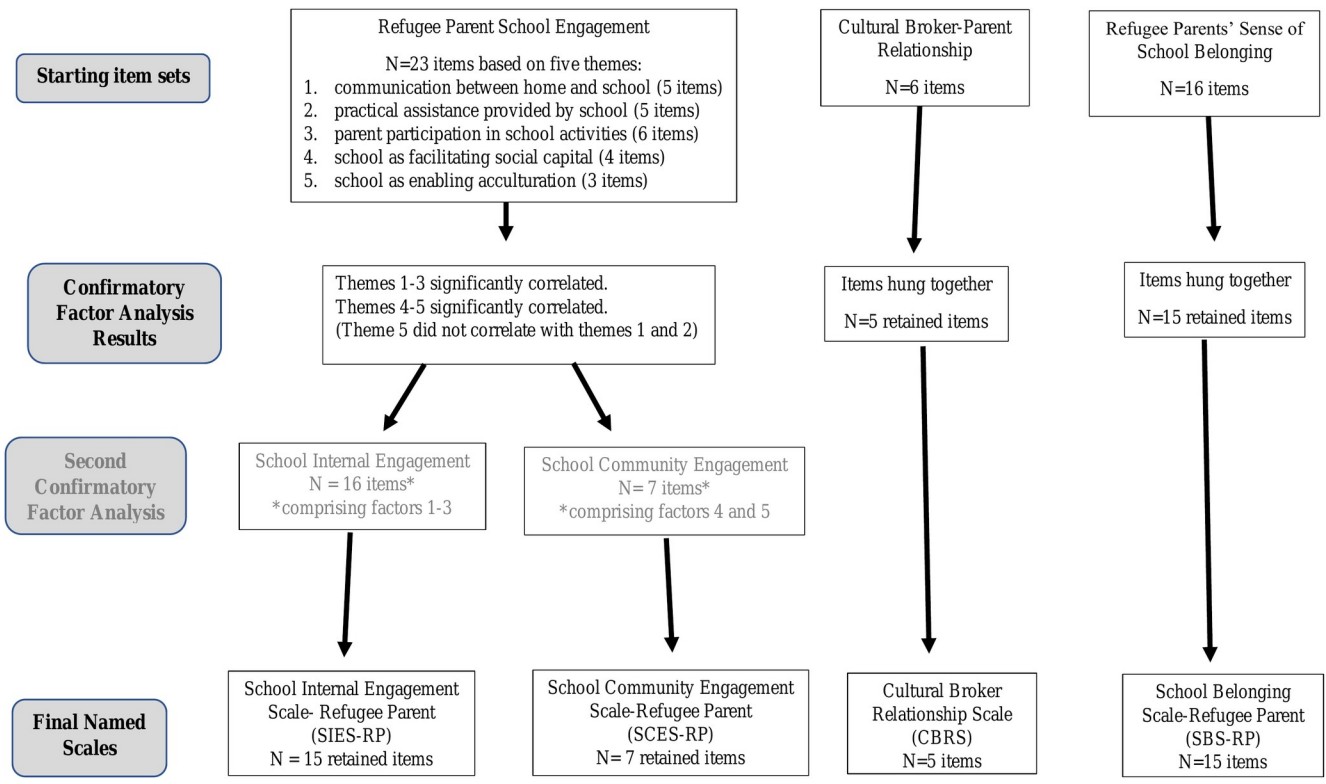

**Fig 2. The process of measure development.**

**Table 2. Values of CFI, TLI, SRMR, AIC, sample size–adjusted BIC, and Cronbach α for each CFA model with number of items included in CFA.**

| Scales measures | Model summary | | | | | Cronbach α |
|---|---|---|---|---|---|---|
| | CFI | TLI | SRMR | AIC | Sample size–adjusted BIC | |
| **SIES-RP** | | | | | | |
| Model 1 (16 items) | 0.47 | 0.38 | 0.09 | 9,971.0 | 9,984.1 | 0.63 |
| Model 2 (11 items) | 0.60 | 0.49 | 0.08 | 6,834.6 | 6,843.6 | 0.67 |
| **SCES-RP** | | | | | | |
| Model 1 (7 items) | 0.84 | 0.75 | 0.06 | 4,519.9 | 4,525.6 | 0.73 |
| **SBS-RP** | | | | | | |
| Model 1 (16 items) | 0.74 | 0.70 | 0.07 | 8,185.7 | 8,199.0 | 0.77 |
| Model 2 (15 items) | 0.76 | 0.72 | 0.07 | 7,435.3 | 7,447.7 | 0.80 |
| **CBRS** | | | | | | |
| Model 1 (6 items) | 0.79 | 0.65 | 0.08 | 2,820.5 | 2,824.3 | 0.53 |
| Model 2 (5 items) | 0.81 | 0.62 | 0.08 | 2,166.1 | 2,169.2 | 0.63 |

AIC, Akaike information criterion; BIC, Bayesian information criterion; CBRS, Cultural Broker Relationship Scale; CFA, confirmatory factor analysis; CFI, comparative fit index; SBS-RP, School Belonging Scale-Refugee Parent; SCES-RP, School Community Engagement Scale-Refugee Parent; SIES-RP, School Internal Engagement Scale-Refugee Parent; SRMR, standardised root mean square residual; TLI, Tucker–Lewis index.

## Predictive analyses with developed measures

The mean of the school measures by parent characteristics are presented in Table 5. Results of the standardised LASSO regression analyses coefficients (Beta, β) for significant predictors are presented in Table 6. Table 7 presents the forward LASSO regression selection summary for each outcome variable. The VIF for the predictor variables ranged from 1.14 to 2.25. Multicollinearity tolerance statistics ranged from 0.44 to 0.87. While some predictor variables were significantly correlated with one another, the maximum value of correlation coefficient r = 0.48 was less than <0.70; the maximum VIF value of 2.25 was less than 10, and the minimum tolerance statistic of 0.44 was greater than 0.10. This indicated that multicollinearity among predictors was sufficiently low to warrant entry into the LASSO regression.

## School Internal Engagement Scale for Refugee Parents (SIES-RP)

Participants with children at schools C and D reported significantly greater internal school engagement as compared to school E (reference school). The LASSO regression analysis revealed that greater parent psychological distress and the longer the time lived in Australia

**Table 3. School scale measures with mean scale score and reliability coefficients Cronbach α.**

| Measures (number of items in the scale and sample size) | Mean total scale score (SD) [range] | Mean of scale score (SD) [range] | Cronbach α |
|---|---|---|---|
| SIES-RP [11 items] (*n* = 231) | 32.6 (5.4) [16–44] | 2.9 (0.5) [1–4] | 0.67 |
| SCES-RP [7 items] (*n* = 231) | 18.1 (4.7) [7–28] | 2.6 (0.7) [1–4] | 0.73 |
| SBS-RP [15 items] (*n* = 232) | 66.7 (6.5) [41–77] | 4.4 (0.4) [1–5] | 0.80 |
| CBRS [5 items] (*n* = 216) | 17.6 (2.5) [5–20] | 3.5 (0.5) [1–4] | 0.63 |
| THC subscale [4 items, parent report] (*n* = 233) | 14.7 (1.6) [7–16] | 3.7 (0.4) [1–4] | 0.69 |

CBRS, Cultural Broker Relationship Scale; SBS-RP, School Belonging Scale-Refugee Parent; SCES-RP, School Community Engagement Scale-Refugee Parent; SIES-RP, School Internal Engagement Scale-Refugee Parent; THC, Teacher–Home Communication.

**Table 4. Correlation matrix of among scale school measures.**

| Measures (number of items in the scale and sample size) | SIES-RP | SCES-RP | SBS-RP | CBRS | THC |
|---|---|---|---|---|---|
| SIES-RP | 1.00 | | | | |
| SCES-RP | 0.30** | 1.00 | | | |
| SBS-RP | 0.53** | 0.22** | 1.00 | | |
| CBRS | 0.16* | 0.01 | 0.28** | 1.00 | |
| THC scale | 0.51** | 0.23** | 0.60** | 0.19** | 1.00 |

*Correlation significant at $p < 0.05$.

**Significant at $p < 0.01$.

CBRS, Cultural Broker Relationship Scale; SBS-RP, School Belonging Scale-Refugee Parent; SCES-RP, School Community Engagement Scale-Refugee Parent; SIES-RP, School Internal Engagement Scale-Refugee Parent; THC, Teacher–Home Communication.

were each related to lower internal school engagement. Parents on a permanent residency or citizenship visa (as opposed to a humanitarian visa) reported greater internal school engagement. Length of time lived in Australia and parent visa category exerted the largest influence on parents' school engagement, followed by the index of psychological distress. The overall model was significant, accounting for 22.3% of the total variance.

### School Community Engagement Scale for Refugee Parents (SCES-RP)

The LASSO regression analysis revealed that speaking a language other than Arabic, being unemployed, and endorsing more PMLDs were significantly associated with lower school community engagement. The overall model accounted for 9.7% of the total variance.

### School Belonging Scale-Refugee Parent (SBS-RP)

The regression analysis revealed that lower level of education, more PMLDs, and greater psychological distress were significantly correlated with lower school belonging. School D had significant influence on parents' higher school belonging. The overall model was significant and accounted for 14.7% of the total variance.

### Cultural Broker Relationship Scale (CBRS)

The regression analysis revealed that parents who had lived less time in Australia, were educated at a high school or diploma level, or who were unemployed reported a significantly greater quality relationship with the school cultural broker, relative to parents who had lived longer in Australia, educated to a degree level, or who were employed. The overall model was significant and accounted for 11.0% of the total variance.

## Discussion

This paper describes the development of 4 contemporary measures of refugee parents' engagement with their child's school, namely the SBS-RP, the CBRS, the SIES-RP, and the SCES-RP. The SBS-RP and SCES-RP demonstrated adequate to good reliability. The reliability of the SIES-RP and CBRS was below the acceptable threshold. All measures indicated sound construct validity, and convergent validity as based on positive correlations with the THC scale. Using these instruments, we studied a cohort of Arabic-speaking refugee parents in Australia. We report that parental factors associated with school engagement included time lived in Australia, psychological distress, PMLDs, visa status, employment status, education level, and speaking Arabic (the dominant language of the sample—69%) at home. Greater psychological

**Table 5. Mean of the school measures by parent characteristics.**

| Parental characteristics | SIES-RP | SCES-RP | SBS-RP | CBRS |
|---|---|---|---|---|
| | Mean (SD) | Mean (SD) | Mean (SD) | Mean (SD) |
| All# | 3.0 (0.5) | 2.6 (0.7) | 4.4 (0.4) | 3.5 (0.5) |
| **Sex** | | | | |
| ˚Male | 3.1 (0.4) | 2.6 (0.6) | 4.5 (0.4) | 3.5 (0.4) |
| ˚Female | 2.9 (0.5) | 2.6 (0.7) | 4.4 (0.4) | 3.5 (0.5) |
| **Marital status** | | | | |
| ˚Married | 3.0 (0.5) | 2.6 (0.7) | 4.5 (0.4) | 3.5 (0.5) |
| ˚Divorced/Separated/Widowed | 2.9 (0.5) | 2.6 (0.8) | 4.4 (0.5) | 3.5 (0.5) |
| **Current parent visa category in Australia** | | | | |
| ˚Humanitarian refugee/Family sponsored | 2.9 (0.5) | 2.6 (0.7) | 4.4 (0.4) | 3.5 (0.5) |
| ˚Permanent resident or Citizen | 3.1 (0.5) | 2.6 (0.7) | 4.5 (0.4) | 3.4 (0.5) |
| **Country of birth** | | | | |
| ˚Iraq | 3.0 (0.5) | 2.6 (0.7) | 4.4 (0.4) | 3.5 (0.4) |
| ˚Syria | 3.0 (0.5) | 2.4 (0.7) | 4.5 (0.3) | 3.6 (0.7) |
| ˚Other | 2.8 (0.5) | 2.4 (0.7) | 4.4 (0.4) | 3.3 (0.6) |
| **Spoken language at home** | | | | |
| ˚Arabic | 3.0 (0.5) | 2.7 (0.7) | 4.4 (0.4) | 3.5 (0.5) |
| ˚Others | 2.9 (0.5) | 2.4 (0.6) | 4.5 (0.4) | 3.6 (0.5) |
| **English proficiency** | | | | |
| ˚A little/Not at all | 2.9 (0.5) | 2.6 (0.7) | 4.4 (0.4) | 3.6 (0.5) |
| ˚Well/Very Well | 3.0 (0.4) | 2.7 (0.7) | 4.4 (0.4) | 3.3 (0.5) |
| **Highest level of education attained** | | | | |
| ˚Up to high school | 2.9 (0.5) | 2.6 (0.6) | 4.5 (0.4) | 3.6 (0.4) |
| ˚Diploma and University degree | 3.0 (0.5) | 2.7 (0.7) | 4.4 (0.5) | 3.4 (0.6) |
| **Employment status** | | | | |
| ˚Unemployed and others | 3.0 (0.5) | 2.6 (0.7) | 4.4 (0.4) | 3.6 (0.5) |
| ˚Employed | 3.1 (0.4) | 3.0 (0.6) | 4.6 (0.3) | 3.1 (0.6) |
| **Child's enrolled school** | | | | |
| ˚School A | 2.9 (0.5) | 2.6 (0.6) | 4.4 (0.5) | 3.6 (0.4) |
| ˚School B | 2.8 (0.5) | 2.4 (0.7) | 4.5 (0.4) | 3.5 (0.7) |
| ˚School C | 3.1 (0.5) | 2.6 (0.7) | 4.4 (0.4) | 3.5 (0.6) |
| ˚School D | 3.2 (0.4) | 2.7 (0.7) | 4.6 (0.3) | 3.6 (0.3) |
| ˚School E | 2.8 (0.5) | 2.6 (0.7) | 4.4 (0.4) | 3.5 (0.5) |

CBRS, Cultural Broker Relationship Scale; SBS-RP, School Belonging Scale-Refugee Parent; SCES-RP, School Community Engagement Scale-Refugee Parent; SIES-RP, School Internal Engagement Scale-Refugee Parent.

distress was implicated in lower parental school engagement across both internal engagement and the more affective sense of belonging. Another factor to reliably emerge as a predictor of school engagement was PMLDs. Specifically, the greater the number of current living difficulties, the lower the parents' reported community school engagement and sense of school belonging. Interestingly, English language proficiency did not show an association with parent school engagement, cultural broker relationship, or sense of belonging. Regarding the cultural broker scale, parents who had lived less time in Australia, were unemployed, or less educated endorsed a greater quality relationship with this school staff member.

It makes sense that the strongest correlations of the 4 self-report tools were between the SBS-RP and the SIES-RP, the 2 measures assessing sense of belonging and internal school

**Table 6. Standardised LASSO (forward) regression coefficients (Beta, β) for significant predictors.**

| Significant predictors# | SIES-RP (n = 231) | SCES-RP (n = 231) | SBS-RP (n = 232) | CBRS (n = 216) |
|---|---|---|---|---|
| | Beta (β) | Beta (β) | Beta (β) | Beta (β) |
| Current visa category in Australia | 0.26** | - | - | - |
| Duration in Australia | −0.24** | - | | −0.24** |
| Language spoken at home | - | −0.15* | | |
| Highest level of education completed | - | - | −0.15* | −0.17** |
| Employment status | - | 0.16** | - | −0.14* |
| Number of PMLDs | - | −0.18** | −0.16* | - |
| Psychological distress (K10 score) | −0.22** | - | −0.23** | - |
| School C# | 0.24** | - | - | - |
| School D# | 0.27** | | 0.16** | |
| *Value of R-square* | *0.241** | *0.118** | *0.166** | *0.124** |
| *Value of adjusted R-square* | *0.223** | *0.097** | *0.147** | *0.110** |

Predictor variables used in LASSO regression analysis are coded as:

Sex: 0 = Male, 1 = Female;

Age (parent): individual age ranges 26 years to 63 years;

Marital status: 0 = Married, 1 = Divorced/Separated/Widowed;

Country of birth: 0 = Iraq, 1 = Syria and others;

Current visa category in Australia: 0 = Humanitarian/Family-sponsored Refugee, 1 = Permanent Resident/Citizen/Others;

Duration of living in Australia: 0 = Under 2 years, 1 = 2 to 5 years, 3 = More than 5 years;

Language spoken at home (parent): 0 = Arabic, 1 = Others;

Level of English efficiency (parent): 0 = A little/Not at all, 1 = Well/Very Well;

Highest level of education attained: 0 = Up to HSC, 1 = Diploma and University degree;

Employment category: 0 = Unemployed/others, 1 = Employed;

Family income (annual): 0 = up to $37,000, 1 = $37,001 and above;

Number of TEs: total counts of premigration traumatic events (0 to 23);

Number of PMLDs: total counts of PMLDs (0 to 25; treated as continuous);

Psychological distress (score): Total score of 10 individual items (K10 score) (10 to 46; treated as continuous);

#For 5 enrolled schools, 4 dichotomous dummy variable created: named as School A, School B, School C, School D where School E considered as reference category.

Outcome variables for each LASSO regression analysis are as follows:

SIES-RP subscale score;

SCES-RP subscale score;

SBS-RP subscale score;

CBRS subscale score.

*Significant at $p < 0.05$.

**Significant at $p < 0.01$.

CBRS, Cultural Broker Relationship Scale; HSC, xxxx; K10, Kessler Psychological Distress Scale; LASSO, least absolute shrinkage and selection operator; PMLD, postmigration living difficulty; SBS-RP, School Belonging Scale-Refugee Parent; SCES-RP, School Community Engagement Scale-Refugee Parent; SIES-RP, School Internal Engagement Scale-Refugee Parent; TE, traumatic event.

engagement (SBS-RP and SIES-RP). The SCES-RP and CBRS encompass community factors outside of the school or independent of the relationship with the teacher or an ability to speak English and thus add more noise to the data.

The finding that parent psychological distress was related to lower internal engagement and sense of belonging also makes sense. A parent facing mental health issues may be less able to engage with their child's school. They may also be harder to engage or feel more disconnected. This latter idea aligns with bidirectional research that family engagement is necessary for schools to be able to reach parents to help improve their and their child's mental health

**Table 7. LASSO regression calculation: Forward selection summary for each of the outcome variable.**

| Step number | Scale and effect (parameter) entered | Number effects in | PRESS | F Value | Pr > F |
|---|---|---|---|---|---|
| | **SIES-RP** | | | | |
| 0 | Intercept | 1 | 52.3 | 0.00 | 1.0000 |
| 1 | Psychological distress score (measured as Kessler 10) | 2 | 49.92 | 12.45 | 0.0005 |
| 2 | School C | 3 | 47.78 | 11.30 | 0.0009 |
| 3 | School D | 4 | 46.67 | 17.05 | <0.0001 |
| 4 | Current visa category in Australia | 5 | 43.73 | 6.70 | 0.0103 |
| 5 | Duration of living in Australia | 6 | 41.68* | 12.59 | 0.0005 |
| | **SCES-RP** | | | | |
| 0 | Intercept | 1 | 97.11 | 0.00 | 1.000 |
| 1 | Number of PMLDs | 2 | 93.57 | 10.44 | 0.0014 |
| 2 | English proficiency | 3 | 91.06 | 7.63 | 0.0062 |
| 3 | Employment status | 4 | 90.13* | 3.99 | 0.0471 |
| | **SBS-RP** | | | | |
| 0 | Intercept | 1 | 38.37 | 0.00 | 1.000 |
| 1 | Psychological distress score (measured as) | 2 | 35.45 | 21.59 | <0.0001 |
| 2 | Number of PMLDs | 3 | 34.80 | 5.83 | 0.0166 |
| 3 | School D | 4 | 34.28 | 4.59 | 0.0333 |
| 4 | Highest level of education | 5 | 33.72* | 5.62 | 0.0186 |
| | **CBRS** | | | | |
| 0 | Intercept | 1 | 49.47 | 0.00 | 1.000 |
| 1 | Duration of living in Australia | 2 | 46.55 | 14.79 | 0.0002 |
| 2 | Highest level of education | 3 | 45.29 | 7.97 | 0.0052 |
| 3 | Employment status | 4 | 44.80* | 4.18 | 0.0423 |

*PRESS, predicted residual sum of squares statistic (leave-one-out cross-validation).

CBRS, Cultural Broker Relationship Scale; LASSO, least absolute shrinkage and selection operator; PMLD, postmigration living difficulty; SBS-RP, School Belonging Scale-Refugee Parent; SCES-RP, School Community Engagement Scale-Refugee Parent; SIES-RP, School Internal Engagement Scale-Refugee Parent.

[60,61]. Lower parental engagement may be one symptom of the more general finding that poor parent mental health, particularly among female caregivers, is a risk factor for negative child outcomes inclusive of refugee and war-affected communities [62–65]. Nearly half of the current parent sample gave indications of a mild to severe mental health disorder. The high rates of mental health issues seen in the refugee population relative to native-born populations may be why psychological distress emerged as a particular risk factor for refugee parents [66].

It is understandable, too, that parents preoccupied with complex life circumstances may have limited capacity to prioritise school engagement. This is consistent with past research documenting a focus on economic survival as a barrier to engagement [19,20,23,24]. It is interesting that PMLD impacted on parents' more affective sense of school belonging. Based on social and cultural capital theories, one might expect stretched resources to have more impact on the more resource-heavy behavioural or cognitive expressions of school engagement. Alternatively, it could be that stressors that challenge one's sense of belonging within the host country more broadly—such as discrimination, communication difficulties, issues with government services, and fears of being sent home—necessarily impinge then on the sense of belonging within the smaller microsystems of that country, including school.

It was noteworthy that English language was not associated with any of the 4 measures of parent engagement. There is extensive literature citing language as a barrier to school engagement in minority populations [40–45]. However, the schools under study were culturally

diverse (over 90% of families did not speak English as a first language), with Arabic being one of the prominent languages. To this end, the schools were familiar with accommodating language needs perhaps more so than typically less culturally diverse schools.

Viewed collectively, the cultural broker scale findings are encouraging evidence that cultural brokers may be assisting target families most in need, i.e., newly arrived families who are not yet employed. This suggests sage use of school capacity. Families who have lived in Australia for longer and are employed and more educated likely have less need to engage with the school cultural broker, given their likely familiarity with the school and capacity to participate in the community.

However, the finding that the longer a parent had lived in Australia was related to lower internal school engagement is curious, especially given the almost opposite relationship in permanent residency or citizenship status (relative to humanitarian visa status) being associated with higher internal school engagement. One might expect a parent who had lived in Australia for longer to be more socially and culturally resourced and thus more able to engage with the school. One potential explanation, similar to the cultural broker reasoning detailed above, is that schools are so geared towards serving newcomers that long-term residents feel less engaged. The relationship between refugee health and time lived in the resettled country does appear to be a nonlinear one. There is some suggestion of a "honeymoon" period whereby refugee health improves significantly upon arrival in the host country, attributed in part to an acute sense of "relief" and an influx of government support, but once this favourable phase passes and government support ceases, refugee health can worsen [67].

The findings carry authenticity when considered that the refugee families were able to be interviewed in their native language. This diverges from many culturally diverse studies that often require English language proficiency, lending itself to a biased, potentially more acculturated, sample. However, the study encompasses many limitations, not least that the reliability of the measures is compromised with the omission of accredited translators and back translation. Some questionnaire items were also double barrelled and could have been interpreted ambiguously.

The most significant limitation of the study is the cross-sectional design. This hinders any casual interpretation and prevents ruling out any other confounding variable underlying the associations found. Moreover, Arabic-speaking families were almost in the majority in the current participating schools, which limits the generalisability of the findings to other schools where refugee families may subsume more of a minority presence or where the host country's language is dominant. Another weakness is that parental engagement was assessed via only one (parent) informant, and there is a noted tendency for parents to inflate self-reports of their school engagement for understandable desirability motives [68]. Furthermore, the authors are careful that this paper does not assume a responsibility on refugee parents to engage, but rather note that parent engagement is a bidirectional relationship between school and the home.

Future research is required to separate out the nuances of each newly developed measure with distinct measures of validity, including predictive and divergent validity. The sociodemographic findings also need to be replicated in larger, longitudinal samples. For example, the number of employed parents in the study was very small, so any interpretation regarding this factor needs to be treated with caution. The small sample sizes across a wide duration of stay from 3 months to 21 years also unfortunately prohibited comparisons between recently arrived and long-term residents in Australia. This would be useful in interpreting the curious finding that the longer a parent had lived in Australia was related to lower internal school engagement.

Limitations of the emerging psychometrics acknowledged, it is hoped that the developed tools will enable a more targeted robust evaluation of the impact of school initiatives on refugee parent engagement, knowing how vital this is in relation to child outcomes. For example,

this paper now provides a measure with which to quantitatively evaluate the quality of the important relationship between the school-based bilingual cultural broker and parent [6,23,26].

The current study forms the baseline data of a larger study that follows parent and child dyads from the last years of primary school across the transition into secondary school. This will enable us to explore the direction and magnitude of change in the different dimensions of parent engagement, and, importantly, meaningful longitudinal associations between different dimensions of parent engagement and child outcomes. Individual item predictor analysis would also be helpful in guiding interventions and pinpointing exactly which tangible dimensions of engagement schools should be focusing on. For instance, during the scale development, it was pertinent that items related to effective communication correlated strongly with parent engagement items, and items related to social capital correlated strongly with acculturation items. Consideration of online parent connectivity in terms of digital literacy, devices, and network may also be a fruitful future endeavour.

Strong family systems are an important protective factor in optimising refugee children's well-being [34]. In terms of enhancing parent school engagement among refugee families, the study findings support the implementation of initiatives that consider the identification and capacity building of parents who are experiencing psychological distress and/or struggling with resettlement stressors. This invites improvements at both government and school levels. Within the school system, a refugee model of engagement that includes the employment of a bilingual cultural broker, in combination with parent outreach programs, is one seemingly positive way through which to provide targeted support to unemployed newly arrived refugee families [6,23,26,31]. At the macro level, the federal system needs to consider policies that minimise the economic and psychosocial challenges that may interfere with refugee parents' capacity to engage with school.

## Supporting information

**S1 Prospective Protocol. Prospective protocol based on approved ethics submission (HC15833).**
(DOCX)

**S1 STROBE Checklist. Strengthening the Reporting of Observational Studies in Epidemiology (STROBE) Checklist.**
(DOCX)

**S1 Data Tables. Pertaining to the development and psychometrics of study scales.**
(DOCX)

## Acknowledgments

The authors would like to thank the supporting schools, all the parents who participated, and research team members, Sajia Faiz and Holya Hassan.

## Author Contributions

**Conceptualization:** Jess R. Baker, Derrick Silove, Susan Rees.

**Data curation:** Deserae Horswood.

**Formal analysis:** Mohammed Mohsin.

**Funding acquisition:** Derrick Silove, Susan Rees.

**Investigation:** Jess R. Baker, Deserae Horswood, Afaf Al-Shammari.

**Methodology:** Jess R. Baker, Derrick Silove, Deserae Horswood.

**Project administration:** Jess R. Baker, Deserae Horswood, Afaf Al-Shammari.

**Resources:** Jess R. Baker, Deserae Horswood, Afaf Al-Shammari.

**Supervision:** Derrick Silove, Valsamma Eapen.

**Writing – original draft:** Jess R. Baker, Mohammed Mohsin.

**Writing – review & editing:** Jess R. Baker, Derrick Silove, Deserae Horswood, Afaf Al-Shammari, Mohammed Mohsin, Susan Rees, Valsamma Eapen.

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
