## [Decision Letter · Decision Letter 0]

26 Apr 2020

Dear Dr. Baker,

Thank you very much for submitting your manuscript "Psychological distress and resettlement stress are associated with lower school engagement among refugee parents" (PMEDICINE-D-19-03661) for consideration at PLOS Medicine. 

[LINK]

In light of these reviews, I am afraid that we will not be able to accept the manuscript for publication in the journal in its current form, but we would like to consider a revised version that addresses the reviewers' and editors' comments. Obviously we cannot make any decision about publication until we have seen the revised manuscript and your response, and we plan to seek re-review by one or more of the reviewers. 

We expect to receive your revised manuscript by May 15 2020 11:59PM. Please email us (plosmedicine@plos.org) if you have any questions or concerns.

We look forward to receiving your revised manuscript. 

Sincerely,

Emma Veitch, PhD

PLOS Medicine

On behalf of Clare Stone, PhD, Acting Chief Editor,

PLOS Medicine

plosmedicine.org

*I understand the paper was submitted for the journal's special Collection on Refugee and Migrant Health; that collection has now launched (https://collections.plos.org/refugee-health-special-issue) so the paper you've submitted is too late for the launch. However, it can still be considered for the journal and if it passes the journal's usual criteria, published; we'd also hope it can subsequently included in the collection's homepage as well. 

*Please revise the title according to PLOS Medicine's style. Your title must be nondeclarative and not a question. It should begin with main concept if possible. Please place the study design in the subtitle (ie, after a colon) - eg, here, "xxyy: cross-sectional study".

*At this stage, we ask that you include a short, non-technical Author Summary of your research to make findings accessible to a wide audience that includes both scientists and non-scientists. The Author Summary should immediately follow the Abstract in your revised manuscript. This text is subject to editorial change and should be distinct from the scientific abstract. Please see our author guidelines for more information: https://journals.plos.org/plosmedicine/s/revising-your-manuscript#loc-author-summary

*Currently, the abstract states that one aim of the work is to "...measure the effectiveness of school-based parent

 engagement initiatives" - I wasn't clear that this was something that the current study design was capable of doing (given that the study doesn't seem designed to assess the effects of an intervention(s) per se - eg, such as using RCT design) - perhaps this can be considered and the explanation of aims framed a bit more clearly?

*Ideally, please rework the in-text referencing style (this should be simple if referencing software is used) to use sequential numbering in square (not round) brackets. Many thanks

*Can you clarify in the Methods section if the study had a prospective protocol or analysis plan? Please state this (either way) early in the Methods section.

*At the moment, figure 1 is named (in the title and also the in-text callouts) as a CONSORT flow diagram. CONSORT is the reporting tool normally used for randomized trials; the actual figure is fine as it is, but I'd suggest renaming the flow diagram of recruitment, this could just simply say (in title and text) "Flow diagram of Parent/Participant recriutment". 

*We would suggest ensuring that the study is reported according to the STROBE guideline (recommended for observational studies - case/control, cohort, and cross-sectional studies), and include the completed STROBE checklist as Supporting Information. Please add the following statement, or similar, to the Methods: "This study is reported as per the Strengthening the Reporting of Observational Studies in Epidemiology (STROBE) guideline (SChecklist)." The STROBE guideline can be found here: http://www.equator-network.org/reporting-guidelines/strobe/. When completing the checklist, please use section and paragraph numbers, rather than page numbers.

Comments from the reviewers:

Reviewer #1: I confine my remarks to statistical aspects of this paper. While the general approach is appropriate, I have a number of issues to resolve before I can recommend publication

Line 337 Why was 85% chosen? This may or may not be a good choice, depending on what it retains and eliminates. Sometimes it is good to have one or two items with a ceiling, as some respondents may respond "correctly" only to that item. ("Correctly" is not quite right, but really means "most positive way")

Line 338-340 Why remove these items before the CFA? The CFA ought to indicate that they are not useful.

Lines 356-358 This approach, known as bivariate screening, cannot be recommended. The resulting multiplle regression will be wrong in several ways: Standard errors will be too small and, as a result, p values will be too low and CIs too narrow. In addition, the parameter estimates will be biased away from 0. It is best to use substantive knowledge to build a model, but if the authors insist on using an automatic method, then LASSO is usable.

Line 358-359 What exactly does this mean? Categorizing continuous variables is a mistake; it leads to an increase in both type I and type II error.. If there are outliers in an IV, they can be dealt with by choice of method (e.g. robust regression).

Line 359=360 Colinearity can occur between more than two variables. What was done then?

Line 361-363 You cannot accept the null, only fail to reject. If the authors want to do equivalence testing, they can, but that would be a separate procedure.

Line 365-367 I don't understand this. What was done?

Peter Flom

Reviewer #2: Review of PMEDICINE-D-19-03661

Psychological distress and resettlement stress are associated with lower school engagement among refugee parents 

The purpose of the paper is to develop culturally-valid measures of school engagement among refugee parents in Australia and to describe the association of school engagement with other sociodemographic characteristics and psychological states of refugee parents. I very much appreciate the careful attention these authors give to establishing measures appropriate to refugee populations. It is important to not impose etic constructs.

Theoretical and Framing Issues

The paper needs more theoretical development. I understand that the qualitative research findings are reserved for another paper. However, there needs to be a discussion of the literature on parental engagement, which is broader and more sophisticated than the authors' suggest with their statement that existing engagement measures focus primarily on behavioral engagement. Several dimensions of parental/family engagement have been identified in addition to behavioral supports of student learning as evidenced by participation (which I agree is the most common). These include perceptions that the school is a welcoming environment, effective communication with school staff, and sharing power an advocacy, among others. The paper needs a review of these existing constructs and describe how the qualitative data showed that additional dimensions of school engagement are relevant for Arab-speaking refugees or that existing measures do not adequately capture the construct for Arab-speaking refugees. The new measures should fill this void, and predictive and discriminant validity established by testing the association with existing engagement measures. This was done to a limited extent with the Delaware scale, but it all seemed a bit random because of the lack of a theoretical framework. 

The items developed for the scales do not follow best practices. They include reverse-coded items. The questions are complex, which makes it difficult to interpret the answers. Some questions combine parental engagement and barriers to parental engagement in a single question. For example, "Despite the language restrictions I feel adequately informed about what happens at the school." This question assumes that parents experience language restrictions, whereas your data show that many do not. If I don't experience language restrictions, how do I answer this question? Other questions are double-barreled. For example, "XX has been important in making me feel secure in the school because XX understands my family's religious and cultural backgrounds." What if XX understands family background but does not make them feel secure in school? These should have been separate questions. 

The scales contain items that on their face seem to measure distinct dimensions of parental engagement. For example, the individual engagement scale includes questions about financial support, parental behavioral involvement, communication, and language barriers. In the larger literature, as noted above, these are treated as separate dimensions of engagement. One of the limitations of the current research on parental engagement, which also suffers from this type of fuzzy measurement, is that it does not lend itself to guiding interventions. Which of these aspects should schools focus on? Which is foundational for the others to be present? Ken Bollen (1990) has an excellent discussion of how highly correlated items may still represent distinct dimensions of community engagement. 

Concurrent validity, as measured by correlations with the THC, needed some discussion.

Finally, the latter part of the paper and the discussion is too causal in nature, even though it is acknowledged that the data are cross-sectional. It presumes a unidirectional association that poor parental mental health reduces family engagement. Research suggests a bidirectional relationship such that that family engagement is necessary for schools to reach parents to improve mental health of both children and parents. See, for example, McNeely et al. (2020; Journal of School Health) and Ellis and colleagues (e.g., Kia-Keating & Ellis, 2007; Clinical Child Psychiatry and Psychology).

Methodological Suggestions

The authors drop variables due to their data distribution or missing data. They did not say which variables were excluded or how many, and I was left wondering if these variables were conceptually important. Fortunately, Mplus can handle variables with missing data. The variables with ceiling effects can be treated as categorical variables in the CFA model. Given that the authors chose CFA over EFA, I suspect that all of the variables used to measure the latent constructs have theoretical importance and should be included if possible.

The authors also drop variables because they have low factor loadings. What are these variables? Do they form a different, also important dimension of social belonging or did they have low factor loadings due to measurement issues (e.g., confusing wording, reverse scoring)? This gets back to the issue of having a theoretical basis for selecting the items and an empirical basis for dropping them.

It would be valuable to test for factor invariance across groups, particularly between early arrivers and long-term residents/citizens in Australia. Not sure if there is enough statistical power, but could the authors explore this? This would be particularly useful given the unexpected finding that duration of time in Australia is associated with lower school engagement. One potential explanation is that the schools are so geared towards serving newcomers that long-term residents feel less engaged.

Presentation Suggestions

Title: Might you consider replacing the term "refugee parents" with "Arabic-speaking refugee parents" in the title. Refugees are an extremely heterogeneous group.

Essential information about some of the scales is missing in the measures section. How many items are on the Delaware school climate survey teacher-home communication subscale and the PMLD scale. Also, I'm not sure it makes sense to report the reliability of the number of trauma exposures because this is a count variable. Measures of reliability assume that all the variables in your scale measure the same underlying construct, which is not the goal in counting the frequency of different traumatic experiences to determine the level of adversity exposure.

Items of the new scales are only reported in the supplemental index, which will make it difficult for readers to access. These should be reported in the main paper.

The supplemental index is confusing. The first table, presenting a correlation matrix, looks really interesting. But you list scales that you never present in the paper. I'm wondering if these scales wouldn't be more appropriate since they seem to distinguish dimensions of engagement more clearly. They also show very interesting findings, which are consistent with qualitative research and would be useful for policy, e.g., that for refugee parents, perceived quality of communication is highly correlated with assistance from the schools. That kind of specificity is needed for schools to know what to do.

The supplemental tables showing item-to-item correlations uses the variable names instead of the variable labels, making it difficult to read. I had to assume the variables were listed in the same order as they were in the prior table. By "estimates from CFA," did you mean the factor loadings? In these tables it would also be interesting to have an additional column reporting the Chronbach's alpha for the scale if the item had been dropped.

Reviewer #3: This is a well-done study guided by theory with community implications. The lack of more details about the translation process is a weakness and should be expanded or listed as a weakness. Also the lack of analysis incorporating school as a cluster was disappointing. A statistician should review and make suggestions.

Other than those criticisms, it is a well-done, well-written study.

[LINK]

---

## [Decision Letter · Decision Letter 1]

18 Aug 2020

Dear Dr. Baker,

Thank you very much for submitting your manuscript "Psychological distress and resettlement stress are associated with lower school engagement among Arabic-speaking refugee parents: A cross-sectional cohort study" (PMEDICINE-D-19-03661R1) for consideration at PLOS Medicine. 

[LINK]

In light of these reviews, I am afraid that we will not be able to accept the manuscript for publication in the journal in its current form, but we would like to consider a revised version that addresses the reviewers' and editors' comments. Obviously we cannot make any decision about publication until we have seen the revised manuscript and your response, and we plan to seek re-review by one or more of the reviewers. 

We expect to receive your revised manuscript by Sep 08 2020 11:59PM. Please email us (plosmedicine@plos.org) if you have any questions or concerns.

We look forward to receiving your revised manuscript. 

Sincerely,

Clare Stone, PhD

Managing Editor 

PLOS Medicine

plosmedicine.org

Unfortunately 2 of the referees continue to have issues with the analysis and revision. Please address all points and note we will only consult once more with the referees.

Comments from the reviewers:

Reviewer #1: The authors responded to my suggestions.

Remaining issues:

The authors wrote

<<<

If a predictor is not found to be significant in bivariate analyses, then it is typically not found to be statistically significant in multiple regression models, in the presence of all other predictors in the model. As expected,

those predictors not found to be statistically significant in bivariate analyses, were not significant in the multiple regression analyses. The variables found to be significant in bivariate analyses are also significant (p<0.05) in the stepwise multiple regression analysis. This consistency of findings indicates that multiple regression analysis is appropriate for our

data.

>>>

This is not always the case and it overemphasizes statistical significance. The results of stepwise methods are wrong. As I said, p values are too low, standard errors too small, parameter estimates biased away from 0. 

Results from stepwise and bivariate analyses should not be presented.

The fact that none switched from significant to non-sig., or vice versa, is not really important.

<<< In SPSS, stepwise forward linear multiple regression analysis technique is similar to LASSO (Least Absolute Shrinkage and Selection Operator) forward technique. Since the term Multiple Regression is more familiar to readers, relative to LASSO, we have stayed with stepwise forward multiple regression as the analysis of choice. (However, if the reviewers feel strongly about this, we are open to change the name to LASSO linear regression (forward).

>>>

It's not a question of the name, but the whole analysis. The results from LASSO should be presented. The fact that it is less familiar isn't really important -- maybe some readers will learn something.

<<<

To be consistent with other dichotomous and categorical predictor variables, we categorised continuous predictor variables. In bivariate analyses, categorisation of continuous

predictor variables is common, e.g. age group, income group, psychological distress, BMI index etc.

>>>

It's certainly common to do this but it is a mistake to do so.

<<<

In the revised manuscript, we have used stepwise forward multiple regression technique. During the estimation process this technique automatically removes the variable by considering the value correlation and collinearity statistics tolerance. We have revised thetext throughout to accommodate this change.

>>>

This is not correct. See e,,g this thread https://stats.stackexchange.com/questions/186081/is-multicollinearity-an-issue-when-doing-stepwise-logistic-regression-using-aic

What happens is that SPSS will make an essentially arbitrary choice among colinear variables. This won't affect prediction, but it can lead to very poor explanation and to wrong conclusions about which variables are important.

Peter Flom

Reviewer #2: Thank you for attending to my concerns about the lack of justification for the study and the lack of a theoretical framework. The revisions to the introduction helped to address these concerns. The rewriting of the discussion also helped. 

Unfortunately, the methodological approach used still does no align with the new theoretical development presented and is still incomplete.

Content Design and Development of New Measures: I found the formatting of this section confusing. I think it is an easy fix. Isn't the determination of these measures the result of the paper? Yet you describe the measures as if they are already validated. I had expected to see a description of your process for organizing the multiple items on the master list into theoretically distinct constructs as laid out in the intro. This is a major part of the methods that you entirely skip over. Was EFA used for this? Another process? Please describe. Also, please provide information on whether the items were originally written in English or Arabic and, regardless, if they were translated and back-translated.

Line 309: What does it mean to selectively sum a scale?

The presentation of results is still confusing. It appears that you proposed a scale and called it SES-RP. You present no empirical or theoretical justification for it being a single scale, nor do you point to it being an extant scale in the existing literature. You then conduct a CFA of these items and group them into five subscales, but do not present the CFA results that led you to that decision. Finally, based on correlation analysis that doesn't appear to treat the scales as latent constructs, you group the five scales into two scales based on what appears to be your interpretation of what makes sense along with the correlation analysis. I get the sense that you are using CFA in an exploratory rather than a confirmatory manner, but there is insufficient documentation. Why did some items have high factor loadings (presumably) when they were part of one subscale but had a statistically insignificant factor loading when they were part of another subscale? Finally, the naming of a subscale as "parent engagement" when all the scales, collectively, are referred to as parent engagement scales is confusing.

A Cronbach's alpha of 0.67 for an 11-item scale is not typically acceptable, as the alpha is a function of both average inter-item correlation and the number of items—11 items is a large scale. This is also reflected in Supplemental Table 5, which shows that some of the items have low correlations with most of the other items and in Table 3's goodness of fit measures. It would be helpful to see estimates Cronbach's alpha with each item deleted one at a time to affirm that all items belong in the scale or to have a theoretical rationale for why these items should remain in the scale.

In short, the development of the scales still needs more empirical and theoretical justification. I cannot tell if the methods used in this version of the paper are sufficient because not all results were presented. 

Finally, I think the authors might have misunderstood the reviewers' request to take into account the clustering of students in schools. We were not asking for school fixed effects in the models (dummy variables for the schools), although the authors' justification that sample size did not allow for that doesn't really make sense (it is four variables). Rather, we were asking that the authors adjust the standard errors of the regression coefficients, which are underestimated due to the fact that the observations within schools are not independent (e.g., families in the same school work with the same cultural broker). The authors have two choices to address this. They can include school fixed effects (one indicator for four of the five schools, with the fifth one serving as the referent group), or they can adjust the standard errors using the complex samples general linear model option in SPSS (or another software). This probably won't change the results substantively, but since all reviewers requested it, I recommend the authors do it.

In addition, the paper uses terms loosely and vaguely, causing the reader to have to work too hard. Here are some examples in the beginning of the paper. Such examples continue throughout even though I do not note them.

The authors switch language between involvement and engagement (note title and first sentence of discussion) without explicitly defining either one. Please choose a single term, define it, and stick to it.

Line 76: take out the word "socio-demographic" and simply refer to "parent characteristics." Psychological distress is not a sociodemographic characteristic.

Lines 81-82: Low variance explained does not mean a set of variables isn't important. I recommend taking out this statement. 

Lines 87-92: The stated conclusions are not based in the findings of this paper. You provide no evidence of effectiveness of cultural brokers in this study.

Lines 95-99: You say the study was done because schools integrate refugee families into communities, but there is nothing in your study about community involvement. You explain the connection well in the intro, but this edited-down version doesn't really make sense. Isn't the purpose to develop appropriate measures of school involvement for program development and evaluation?

Line 109: Casual or causal?

Lines 151-153: Sentence doesn't make sense. What is a refugee field? Current wellbeing is affected by past trauma. Not sure what it means to say it includes past trauma.

Lines 158-159. You say: "Current measures may not capture this." What is "this"? Unclear.

Line 169: What do you mean by "capture this relationship?" Do you mean measure the quality of this relationship? The content of interactions? Unclear.

Line 170: "The development of such measures is important." What measures? Throughout intro need to be very explicit about the constructs you are talking about, keeping the focus narrowly on the constructs you are proposing to measure. There is a tendency in this paper to keep broadening the scope, which ironically weakens the paper by diffusing the focus.

Line 174-175: "Context-specific measures also carry important implications for advancing efforts to improve refugee parents' engagement…" Unclear. What contexts are you talking about?

Line 181-183. Now you are talking about "traditional sense." Please, just use the names of the constructs this refers to, as you've never defined traditional constructs. Similarly, why call the new constructs "extras?" That minimizes their core importance.

Line 191: I suggest not using the word profiling. For U.S. audiences that has very negative connotations and is never something to be recommended.

Line 251: What are generic school engagement scales? You haven't defined these prior to now.

Lines 234-236: This sample size estimation is completely out of context. Statistical power must be estimated in the context of an intended analysis, and you haven't proposed one yet. What sort of coefficients are you referring to for what statistical model? This should be presented later in the data and analysis sections of the methods.

Lines 258-59: Don't need to state that qualitative data was transcribed or the software you used to analyze it. Not relevant or informative for this study. Better to cite the study so reader can find full details. 

Line 264: Need a little more information on the qualitative pilot. 

Reviewer #3: The authors have addressed my concerns.

[LINK]

---

## [Decision Letter · Decision Letter 2]

3 Dec 2020

Dear Dr. Baker,

Thank you very much for re-submitting your manuscript "Psychological distress and resettlement stress are associated with lower school engagement among Arabic-speaking refugee parents: A cross-sectional cohort study" (PMEDICINE-D-19-03661R2) for consideration at PLOS Medicine.

I have discussed the paper with editorial colleagues and our academic editor, and it was also seen again by 2 reviewers. I am pleased to tell you that, provided the remaining editorial and production issues are fully dealt with, we expect to be able to accept the paper for publication in the journal.

[LINK]

Please let me know if you have any questions. Otherwise, we look forward to receiving the revised manuscript shortly. 

Sincerely,

Richard Turner, PhD

rturner@plos.org

Requests from Editors:

In order to comply with PLOS' data policy, https://journals.plos.org/plosmedicine/s/data-availability, we will need to ask that you provide a non-author contact for readers interested in inquiring about access to study data. 

Please remove "are associated with" from your title so that it conforms to PLOS Medicine's style (i.e., non-declarative titles). Please mention the study setting in the title, e.g., "Psychological distress, resettlement stress and lower school engagement among Arabic-speaking refugee parents in Sydney, Australia: a cross-sectional study". 

Please remove the information on funding and competing interests from the title page. In the event of publication, this information will appear in the article metadata via entries in the submission form. 

Please adapt the "Methods and findings" subsection of your abstract to briefly describe the pilot phase of the study also, quoting quantitative details of the findings.

Please include study dates in the abstract; and add a few words to make it clear what the "Kessler10" is. 

Please also quote aggregate demographic details for study participants in the abstract. 

Please restructure the end of the "Methods and findings" subsection of your abstract so that the final sentence begins "Study limitations include ..." or similar. 

Please avoid claims of "the first" and the like, e.g., at line 83. Where needed, please add "to our knowledge". 

Please trim the "Conclusions" subsection of you abstract to no more than 3-4 sentences, and we ask you to focus the text more on the conclusions of the study and their immediate implications. 

Please reformat the "author summary" so that each of the three subsections consists of 3-4 bulleted points, which should generally comprise a single sentence each. 

Please add a heading to the Introduction. 

We ask you to restructure the Discussion section. The first paragraph should provide a summary of the paper's main findings, and there should be a separate discrete paragraph discussing study limitations. 

Please avoid "a couple of" at line 536. 

Please ensure that all p values are quoted exactly, or for smaller values as p<0.001.

Where appropriate, please substitute "sex" for "gender" throughout the paper. 

Please remove footnotes from your text: these can be integrated into the main text at suitable points. Acknowledgements to reviewers can appear in the acknowledgments section. 

Please remove all reference call-outs from subheadings, e.g., at line 272. 

Throughout the text, please remove spaces from the reference call-outs (e.g., ... refugee students [5,6].").

Please revisit your reference list to ensure that all citations contain full access information and comply with journal format. We note that reference 76, for example, appears to lack full access details. 

Please reformat your supplementary material so that the study protocol and STROBE checklist are both separate supplementary documents, individually referred to in the Methods section (e.g., "See S1_STROBE_Checklist"). 

Comments from Reviewers:

*** Reviewer #1: 

The authors have addressed my concerns and I now recommend publication.

Peter Flom

*** Reviewer #2: 

The authors made some of the requested changes. I thank them for that. 

I have some final suggestions.

p. 8. There seems to be a suggestion that the new measures could be used to identify individual parents at risk of low engagement. The authors need to be careful about making that claim because developing tools for monitoring is very different from developing screening tools. Be careful to frame the purpose of the multivariate analysis as identifying risk factors (vs. parents at risk) for low engagement.

p. 12. The description of the measure development is still confusing. This is primarily an organizational suggestion, but I think an important one. You are presenting the analytic strategy in the measures section. In measures section, I would have expected to see a statement describing the theoretical constructs you are testing with CFA and to be given examples of items that you believe measure each construct. This is missing from the methods section entirely. Then the CFA description should go in the analysis section, which you don't have. Consider adding a heading for Data Analysis (like you have for Measures) and putting all subsections starting with Measurement Development under that heading.

p. 16 - 18. All of the information in the section Item Development and Pilot should be put in the Measures section. I thank the authors for adding this important information. This is the information I was expecting to see on page 12. I would move it into the Methods section, since you are describing the methods of your paper, not the results.

Line 412 - you hypothesized 3 domains but said CFA identified 5. Need more context here on what happened. How did you get to five and how did you name them what you named them? And then you go back to four scales. It is impossible to follow your decision-making here. I've stated this same limitation multiple times now. It is starting to feel futile.

Which leads me to my final point. I hope that going forward in their careers, the authors respond less superficially to reviewer comments than they did in the case of his manusript. It is frustrating to keep saying the same changes are needed over and over. I am committed to this topic and the paper had potential, so I stuck it out. The authors would be better served and get their work published more quickly if they responded more fully to reviewer comments. I am not saying that the authors need to change their paper as the reviewer wishes -- a full response could include an explanation why the reviewer is incorrect in their assessment.

***

[LINK]

---

## [Editor Report · Decision Letter 3]

17 Jun 2021

Dear Dr. Baker,

I am writing concerning your manuscript submitted to PLOS Medicine, entitled “Psychological distress, resettlement stress and lower school engagement among Arabic-speaking refugee parents in Sydney, Australia: a cross-sectional cohort study”.

We have now completed our final technical checks and have approved your submission for publication. You will shortly receive a letter of formal acceptance from the editor.

Kind regards,

PLOS Medicine